# Towards soil-transmitted helminths transmission interruption: The impact of diagnostic tools on infection prediction in a low intensity setting in Southern Mozambique

**Berta Grau-Pujol**[1,2,3]\*, **Helena Martí-Soler**[1], **Valdemiro Escola**[2], **Maria Demontis**[4], **Jose Carlos Jamine**[2], **Javier Gandasegui**[5,6], **Osvaldo Muchisse**[2], **Maria Cambra-Pellejà**[5,6], **Anelsio Cossa**[2], **Maria Martinez-Valladares**[5,6], **Charfudin Sacoor**[2], **Lisette Van Lieshout**[4], **Jorge Cano**[7], **Emanuele Giorgi**[8], **Jose Muñoz**[1]

1 Barcelona Institute for Global Health (ISGlobal), Hospital Clínic—University of Barcelona, Barcelona, Spain, 2 Centro de Investigação em Saúde de Manhiça (CISM), Maputo, Mozambique, 3 Mundo Sano Foundation, Buenos Aires, Argentina, 4 Department of Parasitology, Centre of Infectious Diseases, Leiden University Medical Center (LUMC), Leiden, The Netherlands, 5 Instituto de Ganadería de Montaña (CSIC-Universidad de León), Grulleros, León, Spain, 6 Departamento de Sanidad Animal, Facultad de Veterinaria, Universidad de León, Campus de Vegazana, León, Spain, 7 Expanded Special Project for Elimination of NTDs, World Health Organization Regional Office for Africa, Brazzaville, The Republic of the Congo, 8 Centre for Health Informatics, Computing and Statistics, Lancaster Medical School, Faculty of Health and Medicine, Lancaster University, United Kingdom

\* berta.grau@isglobal.org

## Abstract

World Health Organization goals against soil-transmitted helminthiases (STH) are pointing towards seeking their elimination as a public health problem: reducing to less than 2% the proportion of moderate and heavy infections. Some regions are reaching WHO goals, but transmission could rebound if strategies are discontinued without an epidemiological evaluation. For that, sensitive diagnostic methods to detect low intensity infections and localization of ongoing transmission are crucial. In this work, we estimated and compared the STH infection as obtained by different diagnostic methods in a low intensity setting. We conducted a cross-sectional study enrolling 792 participants from a district in Mozambique. Two stool samples from two consecutive days were collected from each participant. Samples were analysed by Telemann, Kato-Katz and qPCR for STH detection. We evaluated diagnostic sensitivity using a composite reference standard. By geostatistical methods, we estimated neighbourhood prevalence of at least one STH infection for each diagnostic method. We used environmental, demographical and socioeconomical indicators to account for any existing spatial heterogeneity in infection. qPCR was the most sensitive technique compared to composite reference standard: 92% (CI: 83%– 97%) for *A. lumbricoides*, 95% (CI: 88%– 98%) for *T. trichiura* and 95% (CI: 91%– 97%) for hookworm. qPCR also estimated the highest neighbourhood prevalences for at least one STH infection in a low intensity setting. While 10% of the neighbourhoods showed a prevalence above 20% when estimating with single Kato-Katz from one stool and Telemann from one stool, 86% of the neighbourhoods had a prevalence above 20% when estimating with qPCR. In low intensity settings,

**Data Availability Statement:** Data cannot be shared publically because of participants consent. But data will be available on a reasonable request to the external data management unit in ISGlobal (ubioesdm@isglobal.es).

**Funding:** BGP and JM received financial support for this study from Mundo Sano Foundation (www.mundosano.org). JG was personally supported at the beginning of the work by the Ramón Areces Foundation and is now funded by the Spanish 'Juan de la Cierva' Programme, Ministry of Economy and Competitiveness (FJC-2018-38305). MMV is personally supported by the Spanish 'Ramón y Cajal' Programme, Ministry of Economy and Competitiveness (RYC-2015-18368). MCP is personally supported by Junta de Castilla y León and Fondo Social Europeo (LE-135-19). ISGlobal is a member of the CERCA Programme, Generalitat de Catalunya. CISM is supported by the Government of Mozambique and the Spanish Agency for International Development (AECID). Prof. Dr. P.C. Flu Foundation also founded this project. The funders had no role in study design, data collection and analysis, decision to publish, or preparation of the manuscript.

**Competing interests:** The authors have declared that no competing interests exist.

STH estimated prevalence of infection may be underestimated if based on Kato-Katz. qPCR diagnosis outperformed the microscopy methods. Thus, implementation of qPCR based predictive maps at STH control and elimination programmes would disclose hidden transmission and facilitate targeted interventions for transmission interruption.

## Author summary

Soil-transmitted helminths (STH) are parasitic worms present in over 1 billion people. Humans release STH eggs to the environment through faeces, and they become infected by egg ingestion or larvae skin penetration. The higher the number of eggs in an infection (high intensity infection) the higher the morbidity severity. For that, the World Health Organization (WHO) aims to eliminate STH as a health problem by reducing moderate and high intensity infections below 2% in 96% of endemic countries by 2030. STH infections main control strategies are mass drug administration to people at risk, and water, sanitation, and hygiene improvements. Nowadays, some regions are reaching WHO goals. Thus, tools to confirm if control strategies could be halted are needed. In this study, we selected a low intensity district in Southern Mozambique where we evaluated how different diagnostic techniques detect STH when intensity of infection is low. In addition, we also created district maps for STH estimated infection prevalence based on the different diagnostics assessed in order to identify the location of ongoing transmission. qPCR showed to be the most sensitive technique. Hence, maps based on the widely used Kato-Katz could underestimate the STH prevalence and lose hidden transmission.

## Introduction

More than 1 billion people have soil-transmitted helminths (STH) infection, worms that develop part of their life cycle in the soil and are transmitted to humans by egg ingestion or skin penetration. They are *Ascaris lumbricoides*, *Trichiuris trichiura*, *Necator americanus* and *Ancylostoma duodenale*. These infections are widely distributed in tropical and subtropical areas, with the greatest burden on children and on poor populations. People with light intensity STH infections can be asymptomatic, but heavier infections can cause anaemia, malnutrition, impaired growth and delayed development, among others [1].

Thus, the World Health Organization 2021–2030 Neglected Tropical Diseases Roadmap has set the ambitious goal of reducing the proportion of moderate and high intensity infections to less than 2% in 70% and 96% of endemic countries by 2025 and 2030, respectively [2]. The mainstay of STH morbidity control strategy is mass drug administration (MDA) with anthelmintics to population at risk, once a year if prevalence is over 20% and twice a year if it is over 50%. To decrease transmission and reinfection, WHO also recommends interventions intended to increase the accessibility to safe water, improved sanitation, and hygiene (WASH) [3].

Some regions are approaching WHO goal of reducing the proportion of moderate and high intensity infections to less than 2% and, thus, reaching STH elimination as a public health problem [4,5]. But transmission could rebound from residual hotspots if interventions are halted without a thorough epidemiological evaluation [6]. Hence, firstly, sensitive and specific diagnostic methods to low intensity infections are essential [7,8]. WHO guidelines recommend detecting and measuring infection with Kato-Katz technique based on a single stool sample

per child [3,9]. If this technique is suitable for low intensity settings is questionable [8]. Secondly, robust survey methods to identify ongoing transmission hotspots are also crucial [7,10]. Cross-sectional surveillance studies can be costly and arduous. But geostatistical modelling showed to be an efficient method to improve surveillance at a reduced cost [11,12]. However, geospatial prediction might be sensitive to the diagnostic technique employed [10].

In Mozambique, a national survey conducted in 2005-2007 showed a STH prevalence of 53.5% in schoolchildren [11]. That prompt the NTDs national program to start yearly MDA in 2011. In 2018, estimated prevalence of moderate-to-heavy intensity infection exceeded the 2% target threshold [13]. Nevertheless, some regions in Mozambique appear to have reached WHO goals.

In this study, we aim to appraise the value of diagnostic methods on estimating the prevalence of infection when assessing ongoing transmission. Therefore, we predicted the prevalence of STH infection based on different diagnostic methods, including the recommended Kato-Katz, in a low intensity area in Mozambique where unsafe WASH conditions abound.

## Methods

### Ethics statement

The study was performed according to the Declaration of Helsinki (version of Fortaleza, Brazil, October 2013), current ICH-GCP guidelines and all applicable national and local regulatory requirements (Spanish Royal Decree 1090/2015). National Bioethics Committee for Health in Mozambique granted approval for this study (Ref.:517/CNBS/17). Participation in this study was voluntary. We obtained written informed consent in either Portuguese or Changana from all study participants older than 18 years all. Caregivers provided written informed consent for participants under 18 years all. Participants between 15 and 17 years old also provided written informed assent. For illiterate caregivers, informed consent was conducted in presence of a literate witness independent from study.

### Study area

The study was conducted in Manhiça District, Southern Mozambique, a rural setting with a predominantly young population [14]. The climate is subtropical with a warm and rainy season (November to April) and a cool and dry season (June to October). The average annual temperatures oscillate from 22˚C to 24˚C and the average annual precipitation from 600mm to 1000mm. Villages typically comprise a loose conglomeration of compounds separated by garden plots and grazing land. Houses are simple, with walls typically made of cane and conventional material. The main occupations are farming, petty trading and employment on a large sugar cane estate.

Since 1996, the Manhiça Health Research Centre (Centro de Investigação em Saúde de Manhiça, CISM) runs a Demographic Surveillance System (DSS) in the area and a morbidity surveillance system at the Manhiça District Hospital and the other peripheral district health posts. The DSS currently covers the entire district: 2380km$^2$ with a population of 201,845 inhabitants under surveillance, 44% of which are <15 years of age. The demographic trends in Manhiça District have been described in detail elsewhere [14].

### Sample size justification

In the absence of updated data on STH prevalence after MDA initiation, we assumed a conservative 50% prevalence in both age groups to ensure a sample size enough to achieve a precision of 5% in the estimation of the 95% CIs regardless of the true prevalence. Thus, the estimated

sample size was 384. We assumed 10% of loss to follow-up or participation rejection in all square area and, due to the needed equal division for regular sampling in the study area this 10% arised approximately 13%. Then, we required 440 participants younger than 15 years old and 440 participants older than 15 years old.

### Study population and selection

A regular sampling design was used to obtain spatial representativeness. A regular sampling design is a uniform sampling methodology specified in advance of data collection that sampling points are selected relative to a sampling origin [15,16]. To design it, we divided the whole study area using a regular grid formed by 440 square areas of 1,750m by 1,750m resolution. One household was selected by defined area, the closest to the centroid, and then one person between 5 to 15 years of age and one person older than 15 years old were randomly selected and invited to take part in the study (S1 Fig). In case of participation declining, the following household closest to the centroid was selected and so forth. Those selected individuals that reported having taken anthelmintics any time during the previous 30 days were excluded. When a household was unoccupied or its head refused to take part in the study, the next closest-to-centroid household was selected.

### Sample collection

Samples were collected between December 2017 and December 2018. Two stool samples were collected from each participant on two consecutive days. A field worker visited candidates' household, invited them to participate and provided written informed consent and study questionnaire when they accepted to participate. The field worker provided a stool collection kit (a sterile 50ml flask, a paper sheet, a wood stick and three pairs of surgical gloves) to each participant and instructed them to ensure they collected adequate quantity and quality of sample. Each participant was requested to provide more than 10g of stool, marked with a line in the flask. Samples were collected on the following morning after recruitment and another stool collection kit was provided for a second stool sample. In case of absence, non-eligibility or unwilling to participate, another candidate within the same age group in the household was selected. Collected samples were kept in a cold box and were transported to CISM for laboratory analysis on the day of collection (Fig 1).

### Laboratory methods

**Microscopy detection.**   Soil-transmitted helminths were diagnosed by the microscopic detection of eggs in stool samples using the Telemann concentration technique [17]. For quantitative assessment of infection, a duplicate Kato-Katz thick smear test from Sterlitech Corporation was used. Each smear was examined within 30 min after preparation. Based on the 41.7 mg template, the number of eggs detected per slide was multiplied by 24 in order to obtain the number of eggs per gram of faeces [18]. As an ongoing quality control, slides were stored in a refrigerator and a ten percent were randomly selected and blindly re-read by a second technician within an hour after preparation. In case of inconsistency, a third re-examination was performed and the data of the third reading was used. After microscopy examination, samples were aliquoted and stored at—80˚C at CISM for forthcoming molecular analysis.

A study member provided parasitological results to all participants. All parasitological positive participants were treated with albendazole at the clinic following the national guidelines. Following treatment, one stool sample per participant was collected at day 21 and this sample was examined according to the same procedure in order to monitor cure [18].

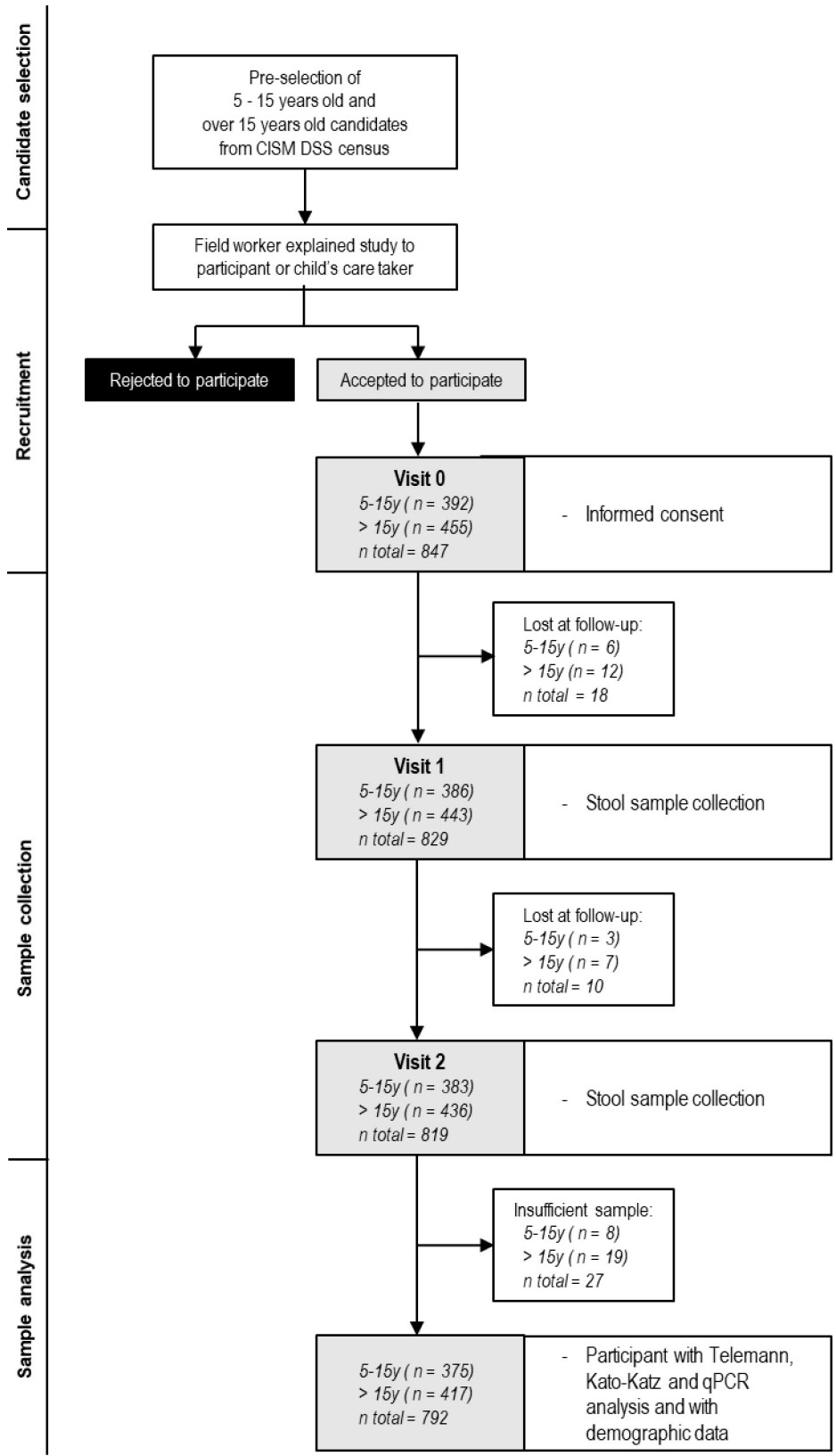

**Fig 1. Diagram flow of recruitment of 5 to 15 years old participants and over 15 years old participants and their sample collection and analysis.**

**DNA extraction and qPCR.** Unpreserved stool samples were stored at—20˚C and transported under frozen conditions to Leon University, Spain. There, DNA extraction was performed using the PowerFecal Pro DNA kit (Qiagen; Germany), following manufacturer's instructions, using 250 mg of stool as input material. The extracted DNA samples were stored at -20˚C and sent to the Leiden University Medical Centre, Netherlands, for further analysis. Two different multiplex real-time PCR detection panels were used to detect and quantify parasite specific DNA as previously published [19–21]. Panel 1 targets *A. duodenale*, *N. americanus*, *A. lumbricoides* and *S. stercoralis* while panel 2 targets *T. trichiura* and *Schistosoma* spp. Amplification, detection and analysis were performed using the CFX real-time detection system (Bio-Rad laboratories). The PCR output consisted of a cycle threshold (Ct) value. Ct values were analysed using Bio-Rad CFX software (Manager V3.1.1517·0823) [21]. Samples run for 50 cycles and no Ct cut-off was used. Further specifications of the used primers and probes, including the sequences and their original publications, have been summarized in S1 Table. Appropiate negative and positive control samples were included in each PCR run. Also the LUMC-team scored 100% for each included target at the annual international helminths external quality assessment scheme (HEMQAS) provided by the Dutch Foundation for Quality Assessment in Medical Laboratories (SKML) [22]. The outcome of *Schistosoma* spp. and *S. stercoralis* PCR will not be presented here.

**Obtaining and processing risk factors data.** Spatial datasets on environmental factors know to be associated with the presence and transmission of STH infections were obtained for analysis [23]. These included: i) mean, minimum and maximum estimates of land surface temperature (LST) [24], ii) average vegetation indices (normalized difference vegetation index (NDVI), enhanced vegetation index (EVI)) and middle-infrared (MIR) [25], iii) top soil pH H20 [26] and, iv) soil composition (clay, silt and sand fraction of the top soil) [26].

Due to the relatively small study area, we resorted to satellite images provided by the Moderate Resolution Imaging Spectroradiometer (MODIS) instrument operating in the Terra spacecraft (NASA), which measure 36 spectral bands and it acquires data at lowest spatial resolution of 250m. From the family of MODIS products, we downloaded global MOD13Q1 and MOD11A2 data, which are provided every 16 days at 250m spatial resolution [24,25]. The MOD13Q1 product includes vegetation indices such as Normalized Difference Vegetation Index (NDVI) and Enhanced Vegetation Index (EVI). The latter minimizes canopy background variations and maintains sensitivity over dense vegetation conditions. It also includes mid-infrared band (MIR) which has been found to be useful to discriminate water surfaces; water highly reflects wavelength in the range of MIR band (2.1 μm) [27]. The MODA11A2 product includes gridded continuous measures of land surface temperature; night (night-LST) and day (day-LST). Fortnightly continuous gridded maps of day-LST, NDVI, EVI and MIR for the study area were produced for 2017 and 2018, and eventually aggregated by calculating the mean, minimum, maximum and standard deviation for the entire period. Soil data including soil-pH and soil composition (fraction of clay, sand and silt) at the top soil, were obtained from the ISRIC-World Soil Information project [26] (S2 Table). This project provides gridded maps of soil composition at 250m resolution worldwide. All mentioned environmental variable processing was conducted using ArcMap 10.5 (ESRI, Redlands, CA).

Population density and socioeconomical, water and sanitation information were obtained from CISM DSS. Population density was estimated by the number of households located in a radius of less than 1km around every household.

A socioeconomical wealth index was built based on household characteristics and assets to attribute a household socioeconomic score (SES) to each household of Manhiça district. For that, we performed a multiple correspondence analysis (MCA) to determine the weights of every characteristic or asset and 17 household characteristics and assets were included [28].

We excluded water and sanitation variables to account them separately, since they are key factors of STH infection. [29] Manhiça district household SES was then divided in tertiles to classify households in poor socioeconomical status, middle socioeconomical status and rich socioeconomical status during participants' description. SES continuous variable was also kept to be used during statistical modelling of infection.

Water accessibility was established as "water inside the household" or "water outside the household" and "water free" or "having to pay for water." Sanitation standard was defined by owing a latrine or toilet in the household or not.

## Data analysis

**Participants' description.** Participant population and STH infections were described using absolute and relative frequency. Categorical variables were expressed as absolute frequency and percentage and they were compared with Chi-square test.

**Diagnostic sensitivity.** We estimated the diagnostic sensitivity for Telemann technique from one stool sample (first sample) and two stool samples (first and second sample), single and duplicate Kato-Katz in one stool sample, single and duplicate Kato-Katz in two stool samples and qPCR. For single Kato-Katz we always used the first slide read in each duplicate Kato-Katz. Due the absence of a STH gold standard diagnostic technique, we defined the diagnostic sensitivity using a composite reference standard (CRS) as a proxy. The CRS was built using all the techniques described above; where a participant was considered infected if positive for at least one of the diagnostic methods used.

**Intensity of infection.** We used WHO thresholds [30] to classify the intensity of infection per parasite based on the Kato-Katz faecal egg count. We evaluated the faecal egg counts agreement (Spearman's rank correlation coefficient) [31] with single Kato-Katz in one stool sample for: duplicate Kato-Katz in one stool sample, single Kato-Katz in two stool samples and duplicate Kato-Katz in two stool samples. Moreover, we evaluated the faecal egg counts agreement (Spearman's rank correlation coefficient) [26] with qPCR in one stool sample for: single and duplicate Kato-Katz in one stool sample, and single and duplicate Kato-Katz in two stool samples.

**Statistical modelling of infection.** All statistical analysis was carried out in R Statistical Software Version 3.5.3 [32] and STATA version 16 (StataCorp., TX, USA). Collinearity was explored using cross-correlations among explanatory variables. To address the issue of collinearity in our model, if a pair of variables had a Pearson's correlation coefficient greater than 0.80, only one of two them was retained in the model. The choice of which variable was kept in the model is explained below.

We predicted household level prevalence of at least one STH infection. For that, we fitted a generalized linear model assuming a binomial distribution (STAT R package) to model Telemann, Kato-Katz and qPCR detected at least one STH infection (positive or negative). The explanatory variables used were inhabitants' age and gender, household water accessibility, owing a latrine, number of households located in a radius of < 1km around each household, top soil sand fraction, top soil pH, LST mean and NDVI mean. (S2 Table) SES was not included in the model since showed high collinearity with water and sanitation variables and environmental variables. Household prevalence was predicted from the response variables: at least one STH infection detected by Telemann (from one or two stools), Kato-Katz (single and duplicate from one and two stools) or qPCR (from one stool) for each participant. For each diagnostic methodology, one, two or four measurements were accounted for each participant considering the number of slides or stools examined. Thus, a participant was positive if at least one of the measurements per methodology was STH positive. All available data was analysed regardless of follow-up condition.

We used variograms (gstat R package) to assess if spatial autocorrelation remains in the residuals (unexplained variance) after fitting the binomial generalized linear model of measured outcome. All variograms of the point predictions built for all diagnostic techniques (Telemann from one stool and from two stools, single Kato-Katz from one stool, duplicate Kato-Katz from two stools and qPCR) were contained within the tolerance limits. Thus, we concluded that the data did not show any strong evidence of spatial correlation(S2 Fig).

We obtained estimated neighbourhood prevalences by averaging household level estimated prevalence per neighbourhood. Thus, we used Manhiça district neighbourhoods to visualize the estimated prevalence for at least one STH infection.

## Results

### Descriptive analysis

We enrolled 792 participants with both microscopy and qPCR results, 375 in the 5–15 years old group and 417 in the over 15 years old. (Fig 1). The group over 15 years old had a highest proportion of women (66.9%) compared to the 5–15 years old group (48.5%, p-value < 0.001).

Around half of the participants (54.7%) had a poor socioeconomical status, a higher proportion in the over 15 years old group (60.9% compared to 47.7%, p-value = 0.001). In terms of water conditions, 32.3% were using unimproved water access, this percentage was slightly higher in the over 15 years old participants (35.3%) than in the 5–15 years old (29.1%, p-value = 0.016). Specifically, 7.7% of all participants used surface water, 9.1% in the over 15 years old and 6.1% in the 5–15 years old group (p-value = 0.016). Regarding sanitation access, no differences were observed between groups: 88.9% were using either unimproved sanitation or open defecation. But a higher proportion of participants owned a latrine in their household in 5 to 15 years old (56.8%) compared to over 15 years old (49.4%, p-value = 0.037). Of those study participants owning a latrine, 7.6% had a hygiene station (soap and water) available at the moment of conducting the questionnaire, 6.6% in 5–15 years old and 8.7% in over 15 years old (p-value = 0.342).

Regarding the number of residents per household, 68.8% lived with more than 4 people; in particular, 73.8% in 5–15 years old participants and 52.3% of over 15 years old participants (p-value < 0.001). Ten per cent in over 15 years old lived alone. (Table 1)

### Prevalence of STH

Manhiça district prevalence for at least one STH by single Kato-Katz in one stool was 13.1%, 11.5% in children between 5 to 15 years old and 14.6% in over 15 years old (p-value = 0.188).

A third of participants (29.8%) were detected to be hookworm infected for at least one of the techniques, followed by *T. trichiura* infected (13.8%) and *A. lumbricoides* infected (8.1%, p-value < 0.001). A higher proportion of participants between 5 to 15 years old than over 15 years old were *T. trichiura* infected (19.2% and 9.1% respectively, p-value < 0.001). In the contrary, a higher proportion of participants in over 15 years old had hookworm infection (38.4% compared to 20.3%, p-value < 0.001). No differences were observed between groups concerning *A. lumbricoides*. qPCR detected the highest number of infections. Regarding hookworm, only *N. americanus* but not *A. duodenale* were detected by qPCR. (Table 2)

qPCR exhibited the highest sensitivity compared to the CRS: 94.9% of sensitivity for hookworm, 94.5% for *T. trichiura* and 92.2% for *A. lumbricoides*. In Kato-Katz, the sensitivity among the number of slides or samples used showed no difference. In Telemann, no difference was observed between using one or two stools neither. For hookworm, Telemann from one stool sample was more sensitive than single Kato-Katz from one stool sample and, Telemann

**Table 1. Description of study participants' characteristics by frequency and percentage, n (%).** Water and sanitation are defined according UNICEF Joint Monitoring Programme [33].

| | Participants 5 to 15 years old (n = 375) | Participants > 15 years old (n = 417) | All participants (n = 792) |
|---|---|---|---|
| **Gender*** | | | |
| Female | 182 (48.5) | 279 (66.9) | 461 (58.2) |
| Male | 193 (51.5) | 138 (33.1) | 331 (41.8) |
| **Age (years old)** | | | |
| 5–10 | 208 (55.5) | | 208 (26.3) |
| 10–15 | 167 (44.5) | | 167 (21.1) |
| 15–24 | | 68 (16.3) | 68 (8.6) |
| 25–34 | | 58 (13.9) | 58 (7.3) |
| 35–44 | | 73 (17.5) | 73 (9.2) |
| 45–54 | | 47 (11.3) | 47 (5.9) |
| 55–64 | | 56 (13.4) | 56 (7.1) |
| 65–74 | | 72 (17.3) | 72 (9.1) |
| ≥ 75 | | 43 (10.3) | 43 (5.4) |
| **Household socioeconomical status*** | | | |
| Rich | 56 (14.9) | 39 (9.4) | 95 (12.0) |
| Middle-class | 140 (37.3) | 124 (29.7) | 264 (33.3) |
| Poor | 179 (47.7) | 254 (60.9) | 433 (54.7) |
| **Main water source access*** | | | |
| water inside the household | 99 (26.4) | 90 (21.6) | 189 (23.9) |
| basic water access | 109 (29.1) | 91 (21.8) | 200 (25.3) |
| limited water access | 35 (9.3) | 51 (12.2) | 86 (10.9) |
| unimproved water access | 109 (29.1) | 147 (35.3) | 256 (32.3) |
| surface water use | 23 (6.1) | 38 (9.1) | 61 (7.7) |
| **Water payment*** | | | |
| payment for water access | 137 (36.5) | 116 (27.8) | 253 (31.9) |
| no payment for water access | 238 (63.5) | 301 (72.2) | 539 (68.1) |
| **Owing a latrine*** | | | |
| household owing a latrine | 213 (56.8) | 206 (49.4) | 419 (52.9) |
| **Main sanitation conditions** | | | 0 (0.0) |
| safely managed sanitation | 30 (8.0) | 22 (5.3) | 52 (6.6) |
| basic sanitation | 19 (5.1) | 16 (3.8) | 35 (4.4) |
| limited sanitation | 1 (0.3) | 0 (0.0) | 1 (0.1) |
| unimproved sanitation | 168 (44.8) | 176 (42.2) | 344 (43.4) |
| open defecation | 157 (41.9) | 203 (48.7) | 360 (45.5) |
| **Handwashing facility** | | | |
| with soap and water | 14 (3.7) | 18 (4.3) | 32 (4.0) |
| without soap or water | 47 (12.5) | 35 (8.4) | 82 (10.4) |
| no facility | 152 (40.5) | 153 (36.7) | 305 (38.5) |
| NA[β] | 162 (43.2) | 211 (50.6) | 373 (47.1) |
| **Total number of household residents*** | | | |
| 1 | 1 (0.3) | 42 (10.1) | 43 (5.4) |
| 2–4 | 97 (25.9) | 157 (37.6) | 254 (32.1) |
| 5–7 | 186 (49.6) | 139 (33.3) | 325 (41.0) |
| 8–10 | 59 (15.7) | 50 (12.0) | 109 (13.8) |
| >10 | 32 (8.5) | 29 (7.0) | 61 (7.7) |

* Chi-square p-value < 0.05.

β NA: Non-applicable. Handwashing facility data was only available for those that had a latrine at home.

**Table 2. Number of participants infected by each soil-transmitted helminth detected by Telemann in one or two stool samples, single and duplicate Kato-Katz in one stool sample, single and duplicate Kato-Katz in two stool samples, multiplex qPCR and by a composite reference standard (CRS, positive for at least one diagnostic technique) per each age group.**

| | 5–15 years old (n = 375) | | | > 15 years old (n = 417) | | | All (n = 792) | | |
|---|---|---|---|---|---|---|---|---|---|
| | *A. lumbricoides* (%) | *T. trichiura* (%) | hookworm (%) | *A. lumbricoides* (%) | *T. trichiura* (%) | hookworm (%) | *A. lumbricoides* (%) | *T. trichiura* (%) | hookworm (%) |
| Telemann x1 [β] | 13 (3.5) | 24 (6.4) | 34 (9.1) | 14 (3.4) | 15 (3.6) | 68 (16.3) | 27 (3.4) | 39 (4.9) | 102 (12.9) |
| Telemann x2 [†] | 18 (4.8) | 31 (8.3) | 44 (11.7) | 19 (4.6) | 20 (4.8) | 86 (20.6) | 37 (4.7) | 51 (6.4) | 130 (16.4) |
| Single Kato-Katz x1 [β] | 12 (3.2) | 19 (5.1) | 21 (5.6) | 10 (2.4) | 13 (3.1) | 48 (11.5) | 22 (2.8) | 32 (4.0) | 69 (8.7) |
| Duplicate Kato-Katz x1 [β] | 13 (3.5) | 25 (6.7) | 26 (6.9) | 12 (2.9) | 14 (3.4) | 55 (13.2) | 25 (3.2) | 39 (4.9) | 81 (10.2) |
| Single Kato-Katz x2 [†] | 17 (4.5) | 27 (7.2) | 32 (8.5) | 14 (3.4) | 18 (4.3) | 68 (16.3) | 31 (3.9) | 45 (5.7) | 100 (12.6) |
| Duplicate Kato-Katz x2 [†] | 18 (4.8) | 32 (8.5) | 38 (10.1) | 16 (3.8) | 20 (4.8) | 76 (18.2) | 34 (4.3) | 52 (6.6) | 114 (14.4) |
| qPCR | 25 (6.7) | 67 (17.9) | 70 (18.7) | 34 (8.2) | 36 (8.6) | 154 (36.9) | 59 (7.5) | 103 (13.0) | 224 (28.3) |
| CRS | 28 (7.5) | 72 (19.2) | 76 (20.3) | 36 (8.6) | 37 (9.1) | 160 (38.4) | 64 (8.1) | 109 (13.8) | 236 (29.8) |

[β] from one stool sample per participant

[†] from two consecutive stool samples per participant

CRS: Composite reference standard, positive for at least one diagnostic technique.

from two consecutive stool samples was more sensitive than single or duplicate Kato-Katz from one stool sample. (Fig 2 and S3 Table)

## STH intensity of infection

Fourteen participants (1.8%) had moderate intensity infection detected by single Kato-Katz from one stool: nine with *A. lumbricoides*, one with *T. trichiura*, three with hookworm and

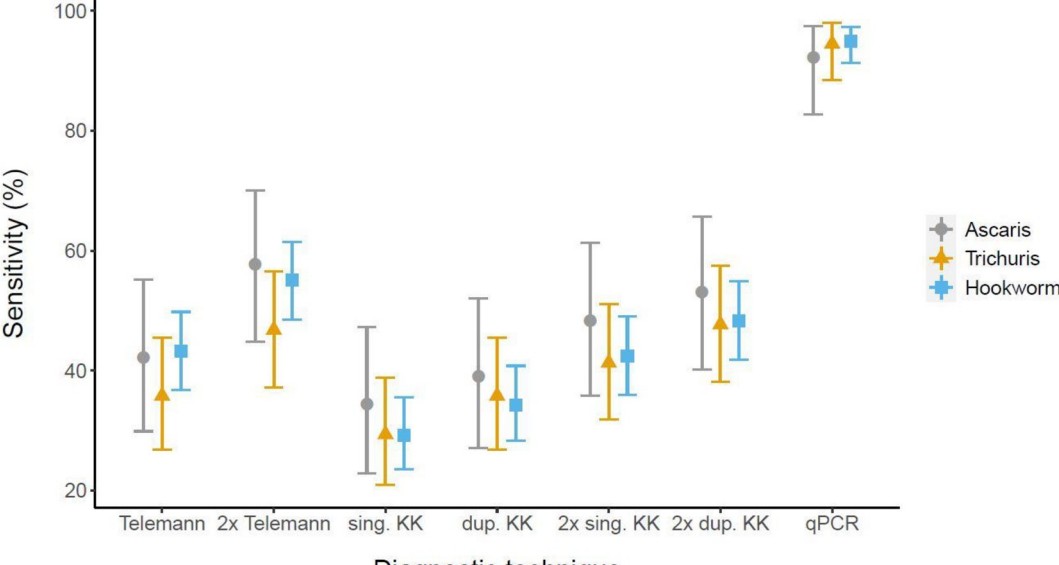

**Fig 2. Estimated sensitivity and 95% confidence intervals [95% CI] of Telemann in one or two stool samples, single and duplicate Kato-Katz in one stool sample, single and duplicate Kato-Katz in two stool samples, multiplex qPCR compared to the composite reference standard (CRS) per *A. lumbricoides*, *T. trichiura* and hookworm.**

**Table 3. Number and percentage of participants with a low, moderate or high intensity of any STH infection detected by single Kato-Katz from one stool; according to WHO. [34].**

| | 5–15 years old (n = 375) | | | > 15 years old (n = 417) | | | All (n = 792) | | |
|---|---|---|---|---|---|---|---|---|---|
| | Low (%) | Moderate (%) | High (%) | Low (%) | Moderate (%) | High (%) | Low (%) | Moderate (%) | High (%) |
| *A. lumbricoides* | 6 (1.6) | 6 (1.6) | 0 (0.0) | 6 (1.6) | 4 (1.0) | 0 (0.0) | 12 (1.5) | 10 (1.3) | 0 (0.0) |
| *T. trichiura* | 17 (4.5) | 2 (0.5) | 0 (0.0) | 13 (3.1) | 0 (0.0) | 0 (0.0) | 30 (3.8) | 2 (0.3) | 0 (0.0) |
| Hookworm | 20 (5.3) | 1 (0.3) | 0 (0.0) | 46 (11.0) | 2 (0.5) | 0 (0.0) | 66 (8.3) | 3 (0.4) | 0 (0.0) |
| At least one STH | 35 (9.3) | 8 (2.1) | 0 (0.0) | 55 (13.2) | 6 (1.4) | 0 (0.0) | 90 (11.4) | 14 (1.8) | 0 (0.0) |

one with *A. lumbricoides* and *T. trichiura* moderate intensity of infection. No participants had high intensity of STH infection. (Table 3)

Regarding Kato-Katz methodologies agreement, the highest agreement with single Kato-Katz from one stool sample was observed in duplicate Kato-Katz from one stool ($\rho > 0.90$) for all STH. In all three species, this was followed by single Kato-Katz from two stool samples and then duplicate Kato-Katz from two stool samples. In the case of *A. lumbricoides*, all three methodologies showed a $\rho > 0.90$. (S3 Fig) In addition, single and duplicate Kato-Katz from two stool samples also displayed good agreement between them ($\rho > 0.90$) in all three species.

qPCR Ct-values and faecal egg counts agreement (single and duplicate Kato-Katz from one and two stools) showed good correlation in *A. lumbricoides*. On the other hand, we observed weak correlation among them for *T. trichiura* and hookworm. (Fig 3)

## District estimated prevalence of STH infection

Neighbourhood estimated STH prevalence based on Telemann from one stool ranged between 1% to 36% and between 5% and 52% from two stools. Ten per cent of neighbourhoods

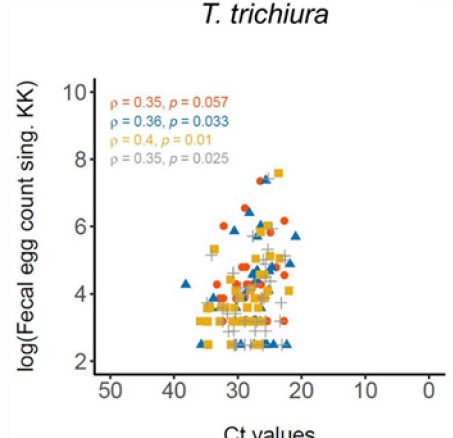
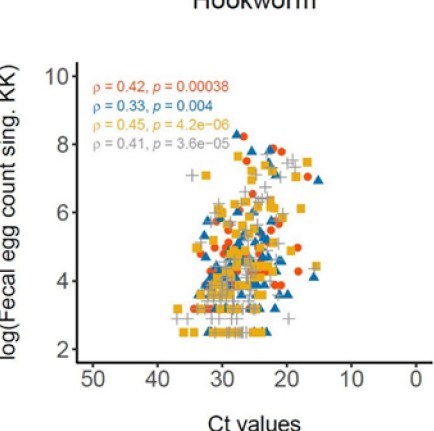

+ single Kato-Katz x1
● duplicate Kato-Katz x1
▲ single Kato-Katz x2
■ duplicate Kato-Katz x2

**Fig 3. Agreement between qPCR Ct value with fecal egg count logarithm of the four quantitative microscopic methods (single and duplicate Kato-Katz in one stool sample, and single and duplicate Kato-Katz in two stool samples) for *A. lumbricoides*, *T. trichiura* and hookworm. In each graph, the concordance correlation coefficient ($\rho$) and p-value are provided.**

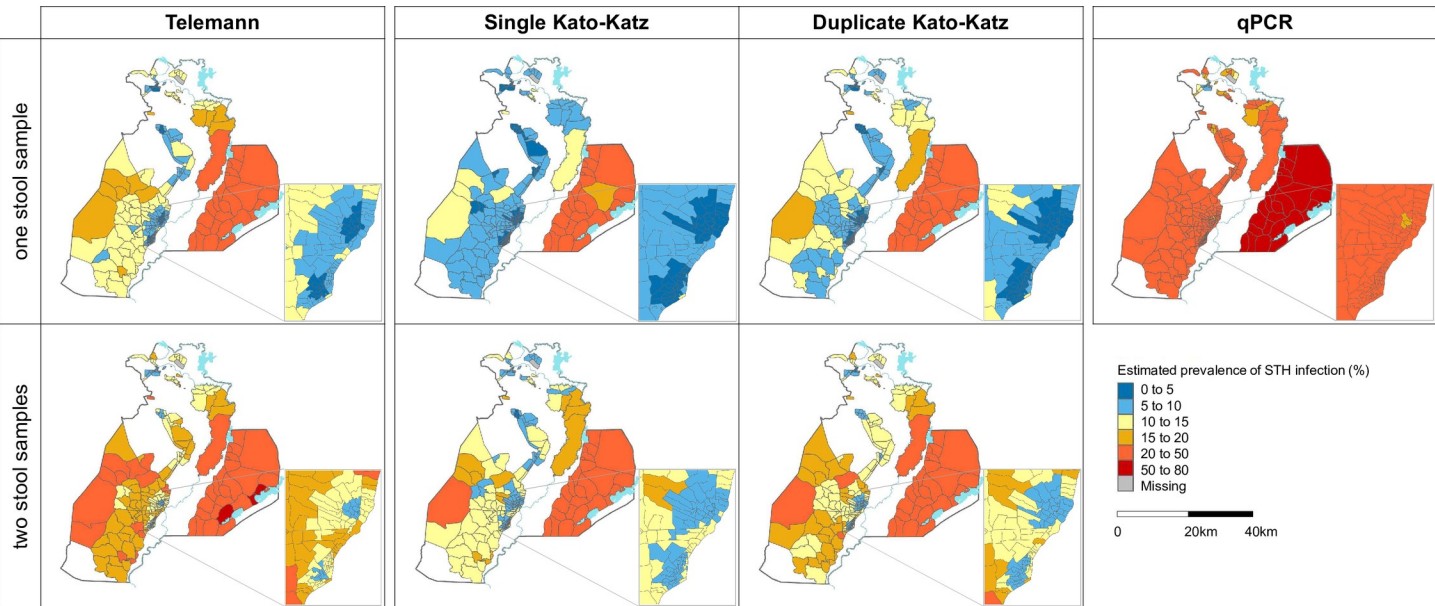

**Fig 4. District map of the estimated prevalence of at least one STH infection calculated with a generalized linear model assuming a binomial distribution for Telemann in one and two stools, single and duplicate Kato-Katz in one stool, single and duplicate Kato-Katz in two stools and qPCR in one stool.** Base layer map obtained in https://data.humdata.org/dataset/mozambique-administrative-levels-0-3.

(n = 25) showed estimated prevalence above 20% when using Telemann from one stool and 16% of neighbourhoods (n = 42) when using Telemann from two stools.

Neighbourhood estimated STH prevalence using single or duplicate Kato-Katz from one stool spanned between 1% and 36% and between 1% and 37% respectively. Estimated prevalence using single or duplicate Kato-Katz from two stools spanned from 4% to 43% and from 4% to 46% in that order. Ten per cent of the neighbourhoods (n = 25) had an estimated STH prevalence above 20% when using single and duplicate Kato-Katz from one stool, 11% (n = 27) of the neighbourhoods based on single Kato-Katz from two stools and 12% (n = 30) of them based on duplicate Kato-Katz from two stools.

qPCR estimated STH infection prevalence per neighbourhood ranged between 6% and 72%. Eighty six per cent of neighbourhoods (n = 222) showed an estimated prevalence above 20% (Figs 4 and S4)

## Discussion

Our results indicate that qPCR is the most sensitive technique and estimated the highest neighbourhood prevalences for at least one STH infection in a low intensity setting. Thus, qPCR should be the preferred diagnostic tool for transmission interruption evaluation.

qPCR detected the highest number of infected participants for all STH species. It also had the highest sensitivity using a composite reference standard; microscopy techniques showed limited sensitivity. This is in accordance with other studies, which expose that microscopy sensitivity is reduced in low intensity infections [8,35–39].

For hookworm, Telemann from one stool had higher sensitivity compared to single Kato-Katz from one stool as well as Telemann from two stools had higher sensitivity than Kato-Katz from one stool (duplicate and single). This could be because hookworm detection by Kato-Katz is very delicate to the time between slide preparation and reading [40–42].

This study did not display statistical evidence of different sensitivity among Kato-Katz number of slides or samples used neither between Telemann number of samples like in other studies [39,43].

Regarding intensity of infection, single Kato-Katz and duplicate Kato-Katz egg count highly agreed, both using one and two stool samples. In this study, samples were thoroughly homogenized before slides were prepared. Slides were read by four different technicians in a blinded way, following good laboratory practice. Furthermore, ten per cent of samples were reread by a senior lab technician. In the laboratory, stool samples were labelled with a unique sample code which could not simply be traced back to the original participant identification number. This in order to prevent any potential bias by the microscopists during the two-slide and two-sample readings. Hence, our results infer that single-slide Kato-Katz would be sufficient if samples are well homogenized, even at low intensities of infection.

We observed a good correlation between faecal egg count and qPCR Ct-values for *A. lumbricoides*, but weak for *T. trichiura* and Hookworm. The reason could be the low intensities of infection–qPCR highly detected infection with very low egg burden. Moreover, there is still little understanding on worm burden and qPCR results relationship [44]. Note that, it is observed that *A. lumbricoides* fertile egg excretion is not linearly correlated with worm burden at low intensity of infections. It is inferred that qPCR could be detecting DNA from other worm specimens than the egg [8,36,45], and thus, detecting prepatent infections (immature worms not releasing eggs) [44]. Other studies found better correlation between eggs per gram and PCR results, but they were using other PCR outcomes such as DNA copies [38,43,46,47].

Geospatial models estimation in Manhiça district allowed us to identify hotspot neighbourhoods for at least one STH infection. This could enable more focused strategies to target areas at risk of continued transmission. [11] In addition, Fronterre et al (2020) showed that geospatial estimated prevalences are a more efficient survey design than classic survey sampling strategies based on random sampling when evaluating transmission interruption. [12]

qPCR model exhibited the highest estimated prevalence of infection of at least one STH per neighbourhood, followed by Telemann from two stools. As mentioned above, qPCR could be informing us of infection and not necessarily morbidity. But, it is necessary to identify infection location—even if low intensity—to target interventions on ongoing transmission sites to achieve transmission interruption goals. If intensity of infection is low, geospatial predictions based on the widely use Kato-Katz could be far from the true prevalence and hidden transmission could still be occurring [3,48]. To solve that, a diagnostic submodel of Kato-Katz egg counting method could be incorporated in the model to account for Kato-Katz poor sensitivity as Farrel et al. 2018 did [49] or estimate with more sensitive diagnostics.

A preferable diagnostic method to monitor transmission interruption could be qPCR. qPCR demands a high-tech lab equipment and high cost consumables [50] but it is very sensitive using a single stool [45]. qPCR could ascertain sustained or increased transmission once MDA is halted [37,38,49]; it was used to confirm transmission interruption in Japan [5]. In addition, qPCR could integrate other helminths surveillance, such as schistosomiasis or strongyloidiasis [51]. A reference lab with appropriate equipment could process the surveillance samples. But before that, some points should be accomplished. Cost reduction is crucial for qPCR implementation. Multiparallel qPCR showed cost diminution [46] and some studies are attempting to optimize a sample pooling protocol to make qPCR surveillance more cost-effective [52]. Moreover, sample storage, DNA extraction and qPCR protocols should be harmonized [45,47,53]. In addition, an external quality assurance for reference labs should be established [49,54]. Lastly, a qPCR threshold based on a universal unit should be created for program decision, to consider patent infections and discard residual DNA detection [44,54].

However, our study has limitations. First, Kato-Katz diagnostic methodologies were compared using nested samples, the first egg count of duplicate Kato-Katz is the same than the single Kato-Katz egg count to which it is compared. Even though microscopy reading bias is unlikely as explained above, some residual bias could have occurred. Second, we fit geospatial models assuming a binomial distribution for all diagnostic methodologies in order to compare them. For low intensity settings, a negative binomial model when estimating STH infection prevalence using quantitative diagnostics could account for overdispersion and add more precision to the model.

Our study observed an area in southern Mozambique where MDA started in 2011 and in 2017, reached WHO guidelines: a prevalence of infection of at least one STH below 20% and had less than 2% moderate to high intensity infections measured by single Kato-Katz. However, water and sanitation conditions were mostly inadequate. Thus, exists a high probability for transmission to rebound if MDA discontinued [55]. Actually, children under 15 years old receiving yearly MDA were still *A. lumbricoides*, *T. trichiura* and/or hookworm infected. Thus, transmission from uncured or adult reservoir may be recurrent.

In this situation, policy-makers should target interventions to ongoing transmission regions regardless of morbidity indicators. STH control programmes need an efficient tool for transmission interruption confirmation, great efforts on that might be cost-saving at long term [7]. This study proposes geospatial prediction as an essential instrument for targeting STH infection areas and discusses the diagnostic implication in this prediction. Community regular sampling (including schoolchildren and adults) in addition to environmental data (e.g satellite images) and socioeconomical data (e.g. DHS) could be used to detect STH infection hotspots at lower sampling costs [12]. The implementation of qPCR based predictive maps at STH control and elimination programmes in low intensity settings would bring programmes closer to the true prevalence and disclose hidden transmission.

## Supporting information

**S1 Table. Oligonucleotide primers and detection probes for multiplex real-time PCR for the simultaneous detection of soil-transmitted helminths.**
(DOCX)

**S2 Table. Potential predictors for human soil-transmitted helminths infection considered for the binomial generalized linear model.**
(DOCX)

**S3 Table. Sensitivity and 95% confidence intervals [95% CI] of Telemann in one or two stool samples, single and duplicate Kato-Katz in one stool sample, single and duplicate Kato-Katz in two stool samples, multiplex quantitative qPCR compared to the composite reference standard (CRS) per for *A. lumbricoides*, *T. trichiura* and hookworm.**
(DOCX)

**S4 Table. Estimated prevalence of at least one STH infection and its associated standard error per region using Telemann from one or two stools, single Kato-Katz from one or two stool samples, duplicate Kato-Katz from one or two stools and qPCR.**
(DOCX)

**S1 Fig.** A) Distribution of all households (black dots) in Manhiça district. B) Household location of study participants (black dots) after regular sampling. Base layer map obtained in
https://gadm.org/download_country_v3.html
(DOCX)

**S2 Fig. Assessment of spatial autocorrelation in the residuals.** Telemann from one or two stools, single Kato-Katz from one or two stool samples, duplicate Kato-Katz from one or two stools and qPCR variograms after fitting the binomial generalized linear model of at least one STH infection.
(DOCX)

**S3 Fig. Fecal egg count agreement of single Kato-Katz in one stool with the other quantitative microscopic methods (duplicate Kato-Katz in one stool sample, and single and duplicate Kato-Katz in two stool samples) for *A*. *lumbricoides*, *T*. *trichiura* and hookworm.** In each graph, the concordance correlation coefficient (ρ) and p-value are provided. The dashed line corresponds to the perfect correlation.
(DOCX)

**S4 Fig. District map of the probability of the estimated prevalence of at least one STH infection exceeding 20% (WHO threshold to start MDA).** It was calculated with a generalized linear model assuming a binomial distribution for Telemann in one and two stools, single and duplicate Kato-Katz in one stool, single and duplicate Kato-Katz in two stools and qPCR in one stool. Base layer map obtained in https://gadm.org/download_country_v3.html
(DOCX)

## Acknowledgments

Our sincere gratitude goes to all field workers and lab technicians that participate in this study for their constancy and persistence. We particularly would like to thank Eric Brienen for his contribution in the qPCR analysis and Llorenç Quintó and Aina Casellas for sample size calculation. We would also like to thank CISM Demography and Social Sciences department for their indispensable assistance. This research was conducted with the support of the Stopping Transmission Of intestinal Parasites (STOP) consortium, that includes: Jose Muñoz, Lisette van Lieshout, Alejandro J. Krolewiecki, Charles Mwandawiro, Rachel Pullan, Inacio Mandomando, Maria Martinez-Valladares, Wendemagegn Enbiale Yeshaneh, Jaime Algorta, Rafael Guille, Nana Aba Williams, Rafael Balaña Fouce, Marc Fernández, Adelaida Sarukhan, Helena Martí-Soler, Berta Grau-Pujol, Javier Gandasegui, Valdemiro Escola, Stella Kepha, Martin Rono, Ellie Baptista, Graham Medley, Catherine Pitt, Augusto Messa Junior, Maria Cambra-Pellejà, and Woyneshet Gelaye.

## Author Contributions

**Conceptualization:** Berta Grau-Pujol, Jose Muñoz.

**Data curation:** Berta Grau-Pujol.

**Formal analysis:** Berta Grau-Pujol.

**Funding acquisition:** Jose Muñoz.

**Investigation:** Berta Grau-Pujol, Valdemiro Escola, Maria Demontis, Jose Carlos Jamine, Osvaldo Muchisse, Maria Cambra-Pellejà.

**Methodology:** Berta Grau-Pujol, Helena Martí-Soler, Javier Gandasegui, Lisette Van Lieshout, Jorge Cano, Emanuele Giorgi, Jose Muñoz.

**Project administration:** Berta Grau-Pujol, Valdemiro Escola.

**Resources:** Berta Grau-Pujol, Anelsio Cossa, Charfudin Sacoor, Lisette Van Lieshout, Emanuele Giorgi, Jose Muñoz.

**Software:** Berta Grau-Pujol, Jorge Cano, Emanuele Giorgi.

**Supervision:** Helena Martí-Soler, Lisette Van Lieshout, Jorge Cano, Emanuele Giorgi, Jose Muñoz.

**Validation:** Helena Martí-Soler, Lisette Van Lieshout, Emanuele Giorgi.

**Visualization:** Berta Grau-Pujol.

**Writing – original draft:** Berta Grau-Pujol.

**Writing – review & editing:** Berta Grau-Pujol, Helena Martí-Soler, Valdemiro Escola, Maria Demontis, Javier Gandasegui, Maria Cambra-Pellejà, Anelsio Cossa, Maria Martinez-Valladares, Charfudin Sacoor, Lisette Van Lieshout, Jorge Cano, Emanuele Giorgi, Jose Muñoz.

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
