## [Decision Letter · Decision Letter 0]

11 Jun 2021

Dear Ms. Grau-Pujol,

Thank you very much for submitting your manuscript "Towards soil-transmitted helminths transmission interruption: the impact of diagnostic tools on infection prediction in a low intensity setting in Southern Mozambique" for consideration at PLOS Neglected Tropical Diseases. As with all papers reviewed by the journal, your manuscript was reviewed by members of the editorial board and by several independent reviewers. In light of the reviews (below this email), we would like to invite the resubmission of a significantly-revised version that takes into account the reviewers' comments. 

We cannot make any decision about publication until we have seen the revised manuscript and your response to the reviewers' comments. Your revised manuscript is also likely to be sent to reviewers for further evaluation.

Sincerely,

Brianna R Beechler, Ph.D., DVM

Associate Editor

Ricardo Fujiwara

Deputy Editor

Reviewer's Responses to Questions

**Key Review Criteria Required for Acceptance?**

**Methods**

-Are the objectives of the study clearly articulated with a clear testable hypothesis stated?

-Is the study design appropriate to address the stated objectives?

-Is the population clearly described and appropriate for the hypothesis being tested?

-Is the sample size sufficient to ensure adequate power to address the hypothesis being tested?

-Were correct statistical analysis used to support conclusions?

-Are there concerns about ethical or regulatory requirements being met?

Reviewer #1: yes

Reviewer #2: -Are the objectives of the study clearly articulated with a clear testable hypothesis stated? yes

-Is the study design appropriate to address the stated objectives? yes

-Is the population clearly described and appropriate for the hypothesis being tested? need to improve

-Is the sample size sufficient to ensure adequate power to address the hypothesis being tested? need to improve

-Were correct statistical analysis used to support conclusions? need more details

-Are there concerns about ethical or regulatory requirements being met? no 

Please see comments for detials.

Reviewer #3: Methods: The field collections performed in this study were extensive and the sampling design appears to be sound. However, more information needs to be included regarding the molecular assays used. The primers and probes used were from previous publications and are referenced in the text. A brief description of the method is still needed: I recommend providing the primer and probe sequences used in a table, as well as the thermocycling protocol used. Also, the method used for quantification needs to be described. This is particularly important as it is compared to FEC later in the manuscript. 

The use of a composite reference standard in this context is also unclear. This method is generally applied to characterize the performance of a new method when there is no "gold standard" reference and in these cases, the composite results of the comparator tests are considered to reflect the "true" disease state. It is generally recognized that this method introduces bias and models that estimate sensitivity and specificity correct for this. The issue here is that the specificity of all three tests are assumed to be 100%. PCR-based tests, while quite sensitive, can also have a high number of false positives, particularly when used for screening in low-prevalence populations.

**Results**

-Does the analysis presented match the analysis plan?

-Are the results clearly and completely presented?

-Are the figures (Tables, Images) of sufficient quality for clarity?

Reviewer #1: yes

Reviewer #2: -Does the analysis presented match the analysis plan? yes

-Are the results clearly and completely presented? yes

-Are the figures (Tables, Images) of sufficient quality for clarity? yes

Please see comments for detials.

Reviewer #3: What is referred to as a composite reference standard in this study seems to actually be a panel diagnostic/screening test. This is fine but more information regarding the specificity of the tests needs to be included. If the authors wish to truly compare the performance of the qPCR panel against the composite reference standard of Telemann and Kato-Katz, more work will need to done to provide a convincing argument, including figuring out how to account for false positives. Overall, the figures are clearly presented

**Conclusions**

-Are the conclusions supported by the data presented?

-Are the limitations of analysis clearly described?

-Do the authors discuss how these data can be helpful to advance our understanding of the topic under study?

-Is public health relevance addressed?

Reviewer #1: n/a

Reviewer #2: -Are the conclusions supported by the data presented? yes

-Are the limitations of analysis clearly described? need to improve

-Do the authors discuss how these data can be helpful to advance our understanding of the topic under study? yes

-Is public health relevance addressed? need to improve

Please see comments for detials.

Reviewer #3: The authors do a good job of describing how the use of highly sensitive tests methods can be incorporated socioeconomic and demographic data can better inform STH disruption strategies. However, the problem of false positive results with qPCR testing needs to be addressed.

**Editorial and Data Presentation Modifications?**

Reviewer #1: (No Response)

Reviewer #2: Requies major revisons

Reviewer #3: I think this study is well done and worthy of publication in NTD once these test-related issues are resolved. One additional note: In a few places in the manuscript "low intensity settings" are mentioned when I think what is meant are low prevalence settings. The terms are often interchanged but it gets confusing when talking about high/low infection intensities (in individuals) and infection prevalence (in a population) in the same paper.

**Summary and General Comments**

Reviewer #1: While the research question is clear and the methods employed are appropriate, this study lacks of novelty. There is a plethora of studies published in the last 10 years that have addressed this research question using more diagnostic techniques and/or in the context of treatment programs (i.e. using follow up samples). Some of these studies have also been referenced by the authors. Please clarify the novelty and usefulness of this study.

Reviewer #2: This study is important as it has evaluated three diagnostic techniques to detect STH infection in a low intensity setting in Southern Mozambique and estimated STH prevalence to identify the locations with ongoing transmission. However, there are a number of issues which require addressing.

Reviewer #3: TH-associated morbidities place severe health burdens on communities in endemic region. The reduction and elimination of STH infections is a critical need and ongoing epidemiological analyses are needed to assess the efficacy of local mitigation efforts. The authors propose a refinement of diagnostic methods to improve identification of STH infections in low prevalence areas by field sampling households in a rural setting in Southern Mozambique. Participants were enrolled and self-collected stool specimens on two separate days. Stools were analysed using two microscopy-based methods (Telemann, Kato-Katz) and two molecular diagnostic (qPCR) panels. The neighborhood-level prevalence was estimated and the resulting map was overlaid with socioeconomical and environmental data to better understand the spatial distribution of STH infections. Overall, I think that this is an interesting and important study but there are some issues that would need to be corrected prior to acceptance.

PLOS authors have the option to publish the peer review history of their article (what does this mean?). If published, this will include your full peer review and any attached files.

Reviewer #1: No

Reviewer #2: No

Reviewer #3: No
---

## [Decision Letter · Decision Letter 1]

11 Aug 2021

Dear Ms. Grau-Pujol,

Thank you very much for submitting your manuscript "Towards soil-transmitted helminths transmission interruption: the impact of diagnostic tools on infection prediction in a low intensity setting in Southern Mozambique" for consideration at PLOS Neglected Tropical Diseases. As with all papers reviewed by the journal, your manuscript was reviewed by members of the editorial board and by several independent reviewers. The reviewers appreciated the attention to an important topic. Based on the reviews, we are likely to accept this manuscript for publication, providing that you modify the manuscript according to the review recommendations. 

Please be sure to consider the concerns about PCR methodology description in your revision. As editor, I agree that a minimum description of the method used improves repeatability and reproducibility and also helps with interpretablity of the outcomes presented here.

Sincerely,

Brianna R Beechler, Ph.D., DVM

Associate Editor

Ricardo Fujiwara

Deputy Editor

Please be sure to consider the concerns about PCR methodology description in your revision. As editor, I agree that a minimum description of the method used improves repeatability and reproducibility and also helps with interpretablity of the outcomes presented here.

Reviewer's Responses to Questions

**Key Review Criteria Required for Acceptance?**

**Methods**

-Are the objectives of the study clearly articulated with a clear testable hypothesis stated?

-Is the study design appropriate to address the stated objectives?

-Is the population clearly described and appropriate for the hypothesis being tested?

-Is the sample size sufficient to ensure adequate power to address the hypothesis being tested?

-Were correct statistical analysis used to support conclusions?

-Are there concerns about ethical or regulatory requirements being met?

Reviewer #2: -Is the study design appropriate to address the stated objectives? yes

-Is the population clearly described and appropriate for the hypothesis being tested? yes

-Is the sample size sufficient to ensure adequate power to address the hypothesis being tested? yes

-Were correct statistical analysis used to support conclusions? need more details

-Are there concerns about ethical or regulatory requirements being met? no

Reviewer #3: I have to disagree regarding the inclusion of minimum details regarding the qPCR assays used. For the sake of reproducibility and the readers' ability to assess the accuracy of the data, the following questions need to be answered in the methods: For how many cycles were the samples run? What was the Ct cutoff used? Was an internal extraction control used? Primer and probe sequences need to be included in the supplement. It is reassuring that the testing laboratory performs annual proficiency testing, however, this additional information is important as the claim of increased prevalence is entirely based on the performance of these assays. 

The sentence defining Ct values is unnecessary. The issue is that the assays were not used to quantify parasite specific DNA in this study. If they were, a description of how the Ct was related to quantification would need to be provided (e.g. plotting against a standard curve). The Ct value itself is not a quantitative measure. In this case, the qPCR assays were used qualitatively.

**Results**

-Does the analysis presented match the analysis plan?

-Are the results clearly and completely presented?

-Are the figures (Tables, Images) of sufficient quality for clarity?

Reviewer #2: Yes

Reviewer #3: (No Response)

**Conclusions**

-Are the conclusions supported by the data presented?

-Are the limitations of analysis clearly described?

-Do the authors discuss how these data can be helpful to advance our understanding of the topic under study?

-Is public health relevance addressed?

Reviewer #2: -Are the limitations of analysis clearly described? need to improve

-Do the authors discuss how these data can be helpful to advance our understanding of the topic

under study? yes

-Is public health relevance addressed? need to improve

Reviewer #3: (No Response)

**Editorial and Data Presentation Modifications?**

Reviewer #2: Minor Revision

Reviewer #3: (No Response)

**Summary and General Comments**

Reviewer #2: This study is important as it has evaluated three diagnostic techniques to detect STH

infection in a low intensity setting in Southern Mozambique and estimated STH prevalence to identify

the locations with ongoing transmission. However, there are a still number of issues which require

addressing. Please refer my comments in the attached.

Reviewer #3: (No Response)

PLOS authors have the option to publish the peer review history of their article (what does this mean?). If published, this will include your full peer review and any attached files.

Reviewer #2: No

Reviewer #3: No

Figure Files:

Data Requirements:

Reproducibility:

References

---

## [Decision Letter · Decision Letter 2]

9 Sep 2021

Dear Ms. Grau-Pujol,

We are pleased to inform you that your manuscript 'Towards soil-transmitted helminths transmission interruption: the impact of diagnostic tools on infection prediction in a low intensity setting in Southern Mozambique' has been provisionally accepted for publication in PLOS Neglected Tropical Diseases.

Best regards,

Brianna R Beechler, Ph.D., DVM

Associate Editor

Ricardo Fujiwara

Deputy Editor

Reviewer's Responses to Questions

**Key Review Criteria Required for Acceptance?**

**Methods**

-Are the objectives of the study clearly articulated with a clear testable hypothesis stated?

-Is the study design appropriate to address the stated objectives?

-Is the population clearly described and appropriate for the hypothesis being tested?

-Is the sample size sufficient to ensure adequate power to address the hypothesis being tested?

-Were correct statistical analysis used to support conclusions?

-Are there concerns about ethical or regulatory requirements being met?

Reviewer #2: Accept

Reviewer #3: (No Response)

**Results**

-Does the analysis presented match the analysis plan?

-Are the results clearly and completely presented?

-Are the figures (Tables, Images) of sufficient quality for clarity?

Reviewer #2: Accept

Reviewer #3: (No Response)

**Conclusions**

-Are the conclusions supported by the data presented?

-Are the limitations of analysis clearly described?

-Do the authors discuss how these data can be helpful to advance our understanding of the topic under study?

-Is public health relevance addressed?

Reviewer #2: Accept

Reviewer #3: (No Response)

**Editorial and Data Presentation Modifications?**

Reviewer #2: Accept

Reviewer #3: (No Response)

**Summary and General Comments**

Reviewer #2: Accept

Reviewer #3: (No Response)

PLOS authors have the option to publish the peer review history of their article (what does this mean?). If published, this will include your full peer review and any attached files.

Reviewer #2: No

Reviewer #3: No

---

## [Editor Report · Acceptance letter]

19 Oct 2021

Dear Ms. Grau-Pujol,

We are delighted to inform you that your manuscript, "Towards soil-transmitted helminths transmission interruption: the impact of diagnostic tools on infection prediction in a low intensity setting in Southern Mozambique," has been formally accepted for publication in PLOS Neglected Tropical Diseases.

Best regards,

Shaden Kamhawi

co-Editor-in-Chief

Paul Brindley

co-Editor-in-Chief
